# Antimicrobial Resistance Research Collaborations in Asia: Challenges and Opportunities to Equitable Partnerships

**DOI:** 10.3390/antibiotics11060755

**Published:** 2022-06-01

**Authors:** Pami Shrestha, Shiying He, Helena Legido-Quigley

**Affiliations:** 1Saw Swee Hock School of Public Health, National University of Singapore and National University Health System, Singapore 117549, Singapore; heshiying@u.nus.edu (S.H.); ephhlq@nus.edu.sg (H.L.-Q.); 2London School of Hygiene and Tropical Medicine, London WC1H 9SH, UK

**Keywords:** antimicrobial resistance, AMR research, collaborative research, equal partnerships, power relations, Asia

## Abstract

Antimicrobial Resistance is recognized as a major threat to global health security. The WHO Southeast Asia region is dubbed a “global hub for AMR emergence”, as it runs the highest risk for AMR emergence among all WHO regions in Asia. Hence, there is a need for Asia-centric, collaborative AMR research aligned with the true needs and priorities of the region. This study aimed to identify and understand the challenges and opportunities for such collaborative endeavors to enhance equitable partnerships. This qualitative study adopted an interpretative approach involving a thematic analysis of 15 semi-structured interviews with AMR experts conducting research in the region. The study identified several factors influencing research collaborations, such as the multi-dimensional nature of AMR, limited or lack of funds, different AMR research priorities in Asian countries, absence of Asia-centric AMR leadership, lack of trust and, unequal power relationships between researchers, and the negative impact of the COVID-19 pandemic in research collaborations. It also identified some opportunities, such as the willingness of researchers to collaborate, the formation of a few networks, and the prioritization by many academics of the One Health paradigm for framing AMR research. Participants reported that the initiation of stronger cross-discipline and cross-country networks, the development of Asia-centric AMR leadership, flexible research agendas with shared priorities, transparent and transferable funds, and support to enhance research capacity in LMICs could assist in developing more equitable collaborative research in Asia.

## 1. Introduction

Known as a silent epidemic spreading insidiously throughout the globe, antimicrobial resistance (AMR) undermines advancements made in modern medicine and presents a critical threat to global health security. AMR refers to the development of resistance to anti-bacterial, anti-viral, and other medicines used against pathogenic infections among microbes [1].

The scale and complexity of AMR demand cohesive and targeted international cooperation. In 2015, the World Health Organization (WHO) launched its Global Action Plan on Antimicrobial Resistance to coordinate key mitigation strategies for AMR (GAP-AMR) across all member states [2]. In the same year, the Global Health Security Agenda (GHSA) was launched to coordinate global efforts to “prevent, detect, and respond to infectious disease threats”, with AMR identified as a key challenge [3]. Indicative of the severity of AMR to global health security, a resolution to combat AMR was also passed in the United Nations General Assembly in the following year [4]. To illustrate the immensity of the AMR threat, a 2014 economic forecast predicted that AMR in six important pathogens will cause an additional 10 million deaths annually by 2050 [5]. This devastation is not discounting the economic cost of USD 100 trillion [5]. Should no further appropriate actions be taken to combat AMR, 24 million additional people are estimated to be forced into extreme poverty, with low-to-middle income countries (LMICs) feeling the brunt of the impact [6]. These numbers should trigger a global sense of urgency, for drug-resistant pathogens already contributing to enormous health burdens around the globe. As of 2014, 3.7 million to 6.4 million bloodstream infections and 28.9 million to 50.1 million serious infections were estimated to be caused by third-generation cephalosporin-resistant *Escherichia coli* and *Klebsiella pneumonia* [7].

### 1.1. Update on the Current Situation in Asia

Combating AMR in Asia is of utmost urgency. The region is home to two-thirds of the world’s population and ten of the world’s least developed countries [7]. Given that the region comprises both LMICs and high-income countries (HICs)—signifying a large variance in levels of AMR drivers and AMR mitigation capabilities across the region—combatting AMR in Asia would be highly complex. For instance, while there were significant gaps in the data on the origin and epidemiology of many AMR organisms and their related outbreaks in Asia, there was already evidence of regional transmission of AMR. A study suggested the presence of regional transmission of multi-drug-resistant (MDR) and hypervirulent *Klebsiella pneumoniae* strains between Cambodia, Thailand, India, Hong Kong, Laos, and Nepal, which was responsible for severe bloodstream infections [8]. Similarly, a separate study also recognized the high possibility of carbapenemase positive *Klebsiella pneumonia* isolates found in Singapore to have originated in China [9]. In light of the gravity of insufficient surveillance data, alongside studies on AMR research and policy implementation in the region, many calls have been made for collaborative efforts on AMR surveillance, data generation, and evidence-based research [10].

A systematic analysis of AMR National Action Plans (NAPs) of ten countries in the region had found that all NAPs emphasized the importance of research activity to understand the drivers and impact of AMR, particularly the knowledge, attitudes, and practices of all stakeholders involved. Five NAPs in Southeast Asia had specifically outlined the importance of research collaborations, particularly through the establishment of research priorities across disciplines and through the engagement of all national stakeholders [11]. Although a declaration was made by the leaders of the Association of Southeast Asian Nations (ASEAN) in 2017 to combat AMR with joint efforts and the One Health approach, the inadequacy of actions to manage AMR and the need for additional multi-sectoral collaboration were acknowledged [12]. Hence, the region requires a better understanding of the importance of targeted collaborative AMR research to inform policy decisions and implementations in combatting AMR [7].

The urgent need to combat AMR in Asia has seen an increase in funding and attention channeled towards AMR research in the region [13]. In 2020, the AMR Action Fund was launched jointly by the WHO, European Investment Bank, and Welcome Trust, totaling USD 1 billion in investments to fund antibiotic development [14]. A 2017 study that aimed to map total investments in AMR research found that approximately 1.8 billion euros were dedicated to AMR research in the European Union (EU) alone, with more expected to follow as EU nations commit to increasing funding in AMR research [15].

### 1.2. Why Collaboration Is Required

There have been calls to coordinate research on an international level to identify research priorities, increase research funding, and focus research activity [16]. Collaboration enables the pooling of resources and harmonization of research activities among states, which can help overcome the limits of nationally funded research programs [15,17]. Further, collaborations present opportunities to conduct research across national borders, which is essential given the transnational nature of AMR [17,18]. Even more pressing is the need for regional collaboration given that current research activities may not be truly aligned with the true priorities of the region [19]. National research activities in Southeast Asia are fundamentally driven by “policy-level prioritization and availability of funding”, which implies the neglect of filling key research gaps in other areas [19]. A review of country situational analysis reports found that although most countries in the WHO South-East Asia Region (SEAR) had banned the selling of antibiotics over the counter, it was still widely practiced in all countries, reflecting the insufficient implementation of legislation aimed to reduce inappropriate antibiotic use [20]. In another instance, a review of policy interventions in the livestock industry among the WHO SEAR had found that although the region had begun to develop policies that are aligned with international standards, such as the WHO Codex Alimentarius to mitigate AMR, efforts were inconsistent across countries [21]. For example, Sri Lanka and the Maldives had passed legislation that banned the use of all antibiotics as growth promoters and their use in medicated animal feed, whereas Indonesia and Thailand had banned the use of antibiotics as growth promoters for animal farming, but only prohibited their use in medicated animal feed in aquaculture. Both studies found that the resources available were insufficient to appropriately implement the policies recommended to combat AMR, and the pooling of resources across countries could solve this problem [20,22]. Given the evident lack of coordination between countries, there is a call for regional collaboration to jointly conduct research to generate evidence identifying the barriers that impede legislature enforcement locally.

Although AMR management is guided primarily by ministries of health at the national level, the transnational nature of AMR demands a cross-country, multidisciplinary, coordinated One Health approach for appropriate AMR research in the region [19]. However, collaboration, particularly at the international level, is easier said than done. Scholars have found that collaboration can be impeded by differences in sociodemographic backgrounds, race, and national origin, and particularly relevant for international collaboration—distance and language differences [23]. Possibly increasing the difficulty of this endeavor, the region’s plans for international collaboration had been identified as lacking [11]. A study had also found that there is a critical need to strengthen healthcare providers’ professional skills in effectively collaborating across disciplines—between doctors, pharmacists, and others in LMICs—signifying that there are already existing barriers to collaboration even at the local level [24].

Given the importance of international collaboration in combating AMR in the region, it is critical to investigate the challenges that impede collaborative research efforts, as well as identify any opportunities that can be leveraged to advance Asia’s research endeavors.

## 2. Method

This qualitative study adopted an interpretative approach to understanding the current situation of AMR research in Asia. Semi-structured interviews were conducted to explore the perception of the AMR experts on the challenges and opportunities of conducting equitable collaborative research. The question guide was developed based on the study’s aim to identify the barriers and opportunities when it comes to AMR research in Asia. Questions were also designed based on study objectives that included the determination of how research priorities may differ between interviewees, the presence of current barriers to research collaborations, and the possible opportunities available for increased collaboration. Finally, questions on concepts related to collaboration, such as power relations, capacity building, and information flow, were also included to obtain a more holistic picture of collaborative efforts in the region. We used purposive sampling to identify AMR experts who were either from institutions based in Asia or have been collaborating with researchers based in Asia. The participants had to be involved in some aspects of AMR research, including but not limited to AMR surveillance, stewardship, public education, infection prevention, or control or research and development. The participants should also have conducted research at the local, national, or global level and be in a capacity to contribute to AMR’s policy-making process. Two landmark documents identified to be important in the AMR discourse in Asia were utilized to identify participants with expertise in AMR. Authors from the Lancet series 2015 titled “Exploring the evidence base for national and regional policy interventions to combat resistance” were identified to fit our recruitment criteria of AMR researchers with critical experience in Asian countries [25]. Actors contributing to the article titled “State of Play of Antimicrobial Resistance Research and Surveillance in Southeast Asia” [19] were also included in our sample list as it was compiled by key stakeholders, such as the ministries, policymakers, research funders, and researchers focused on AMR in the region. As a part of the purposive sampling process, AMR experts in the region who met the inclusion criteria and were not included in the two reports from the regions were also identified through personal contacts. This study received an exemption for ethical approval from the National University Singapore (NUS) Institutional Review Board (IRB) for Social, Behavioural, and Educational Research (SBER).

## 3. Data Collection

A total of 15 participants were interviewed between July 2020 and December 2021. All participants had senior positions at relevant organizations or institutions and were experienced in the field of AMR research in Asia. Considering the limited number of researchers performing cross-border collaborative work in the region, data collection was halted when thematic saturation was reached with a total of fifteen interviews. Interviews were conducted in English via Zoom or Skype which spanned a duration ranging between 30 and 60 min. The participants were contacted via email and sent study information and a consent form. Due to the potential sensitivity of interviews relevant to policy-making processes, we anticipated the possibility that participants would prefer to have their interviews fully anonymized and hence avoided having their informed consent documented. The research team hence obtained only verbal consent from the interviewee which was recorded during the interviews. Participants were ensured total confidentially, with an assurance of being quoted anonymously without reference to their age, sex, professional status, affiliation, and role. Further participants were also provided with the option of not being quoted at all, even anonymously, to refuse to answer any question and/or withdrawing from the interview at any time.

The question guide focused on perceived AMR research priorities in Asia, challenges and opportunities for collaboration in AMR research in Asia, and areas of improvement to develop capacity for AMR research, as well as the impact of the COVID-19 pandemic on AMR research. Table 1 shows the summary of the interview guide. The detailed topic guide for the interview is provided in Appendix B.

## 4. Data Analysis

The interviews were audiotaped and transcribed verbatim and QRS NVivo, a qualitative data analysis software package was used for data management [26]. The study was fully anonymized with no personal data appearing in the interview transcripts. During analysis, each participant was given an identifier code and only this code was attached to the interview transcripts. Participants were quoted without any reference to their affiliation or other personal data.

The data was interpreted inductively using thematic analysis to elicit new themes or unexpected findings through coding of data. The identified themes were supported by the excerpts from the interviews to ensure the data interpretation was true to the opinions provided by the participants. Thematic analysis involved mainly four components: Familiarizing of data; Identifying codes and themes; Line by line coding the data; Organizing codes and themes. We summarized and organized the data based on the themes that emerged. Data collection was ceased when thematic saturation was reached, and new data collected did not add additional insights about the themes identified. Findings were reported according to the COREQ checklist (Appendix A).

## 5. Results

A total of 15 AMR experts participated in our study. These participants represented various countries in Asia, such as China, Indonesia, Japan, Malaysia, Pakistan, Philippines, Singapore, and Thailand. We did not record any further personal details to maintain the anonymity and confidentially of the study participants.

We present our findings under ten main themes on AMR research collaboration, seven are considered challenges, and three themes were reported opportunities for equitable collaboration in AMR research in the region.

The first theme mentioned by participants as a challenge outlines the difficulty of cross-discipline and/or multi-sectoral collaboration due to the multi-dimensional, multi-faceted nature of AMR. The second theme highlights the lack of funding for such collaborations. The third theme expands on the challenge of collaboration due to the differing nature of AMR problems and research requisites in Asian countries. The fourth theme discusses the lack of leadership and Asia-centric institutions to forge the necessary collaborations. The impediment of trust among researchers due to their previous unfavorable experience of unequitable collaborations is the fifth theme. The sixth theme pertains to the unequal power balance among the researchers from countries of different affluence in the region with a tendency among HICs to lead the studies. The seventh and final theme for challenges is the temporary setback caused by the COVID-19 with all healthcare resources being diverted from AMR research to the management and response of the pandemic.

AMR researchers’ appreciation of the need for collaborative work for AMR control and willingness to work together is the first theme for opportunities for such joint ventures. The second theme for opportunity is the presence and formation of a few new networks for communication among AMR researchers. Prioritization of the One Health Paradigm for research in the region, which recognizes the interconnectedness of the human, animal, and environmental domains is the third and final theme for research collaboration opportunities.

We appreciate that more themes for challenges were identified than for opportunities for equitable collaborative AMR research.

## 6. Challenges

### 6.1. Multi-Dimensional, Multi-Faceted Nature of AMR

AMR research often requires the engagement of professionals across various domains of healthcare. Since AMR spans disciplines ranging from microbiology, clinical medicine, environmental, and veterinary sciences to the social sciences and more, differences between research approaches and best practices often emerge. The differences in discipline-based research practices can often result in tensions, which contribute to challenges in streamlining research protocols, aligning their understanding of AMR, or even networking with one another. As a result, researchers mentioned they preferred to partner with similar professionals or within the same disciplinary background. For example, the following quotes illustrate that researchers are influenced by their education and professional culture to associate and form networks within their discipline or health domain:

“Working with similar research background only. I work on molecular pathway; I would probably work with molecular scientist. So that’s sort of an obvious one. The second one is what I just said before. People carry their education and professional culture habits with them, which means they will be prone to collaborate (with similar professions).” (Participant 11)

“Yeah, because… first you have to have the AMR network as the umbrella AMR. And under that, for sure, you have to collaborate through that, that different dimensions, because in some way, in dimension is like, how to say, like for animal size, they would, they would have their, their own network. But anyway, they have to connect with us from human side as well.” (Participant 9)

Researchers have failed to understand the interconnected roles of the various human health domains for AMR control, leading to absence of incorporation of knowledge and perspectives from other health-disciplines during AMR research. The following quote is an example how such misconception prohibits researchers to combine AMR management efforts across the different AMR research domains:

“They still think that the antimicrobial resistance problem or how they can use antibiotic or antimicrobial usage problem is not connected between each other, you know. So they start to make plans separately, which is impossible, you know, because they should be together and to work together. Of course the doctors may think, ‘Oh, I have the patient. I have the decision. I know my patient. I know what to give to my patient.’ But the team from microbiology might think, ‘No, that’s not correct what you give because we have this data.’” (Participant 10)

As part of the complex nature of the AMR problem, participants had raised another notable area of concern, which was the lack of studies on antimicrobial consumption in animal health. Participants also highlighted the current situation of poor knowledge and understanding of the relationships between human and animal health, as well as that with the environment in the region. This posed a challenge for collaborative research between these different domains. The following excerpts illustrate the inconsistencies in current AMR management and studies on antimicrobial use among humans and animals: 

“The level of surveillance of AMR and antibiotic use in the human health sector is very much advanced compared to the veterinary health sectors. The veterinary health sector is just beginning.” (Participant 5)

“So I would say in the Southeast Asian region, I think a big gap is understanding the link between antibiotic use in humans and animals. What is the link with the environment, that’s kind of an area that I’m interested in personally where there’s relatively little work being done? Aquaculture is a huge, huge area is the fastest growing form of agriculture and I think has very strong links to development. There’s very little known about antibiotic use in aquaculture, although people suspect there’s a lot of it being used. So I guess for me those are priorities in the sense that they’re areas that haven’t really been looked at very much.” (Participant 2)

AMR research also requires the exploration of factors beyond public health and the medical domain. AMR issues affect the economy and trade as well as the social structure of the country. Management of AMR requires appropriate policies for use of antibiotics in both human and animal health as well as for farming and production of animal food. Here a researcher described AMR as a “complex issue” and found navigating the social, economic, and political realms of AMR to be difficult due to the difference in priorities of these different collaborators.

“I feel that we struggle a little bit because I think it is such a complex issue. For me it is more of a societal and social structural issue. So I think one of the things that’s really interesting about AMR is that it’s on the kind of biological scales with these kind of tiny you know, changes that happen at the genetic level that have huge social implications.” (Participant 2)

### 6.2. Funding Challenges

Participants reported the lack of or limited funding available for AMR research in Asia as a big hurdle for conducting collaborative studies in the region. Unlike other healthcare crises, such as HIV or COVID-19, AMR is not associated with the same severity or graveness despite its potentially devastating consequences. The next few quotes illustrate that the researchers have only recently initiated AMR research as it has been a neglected and not as popular a topic as the ongoing pandemic:

“So this is the first step of our, yeah, involvement in AMR because, at this moment, AMR is a small, neglected… what do you call neglected side, neglected services at this moment.” (Participant 6)

“I think I think one thing, one more important thing is AMR is not popular like COVID. No. Because COVID-19 is so popular and then it’s easily to detect, easily to find prognosis.” (Participant 7)

For this reason, participants believe that funding for AMR research is not being prioritized by funding bodies. The following quote illustrates how it is difficult to sustain collaborative work without appropriate funds:

“I think it’s improved in the sense of being able to form collaborations. I think it’s difficult to sustain them just because the funding is not there.” (Participant 2)

Furthermore, pharmaceutical companies found AMR research financially unattractive as the development of newer drugs requires a significant length of time. There is also a high probability of the emergence of new strains of resistant organisms, rendering the antimicrobials ineffective by the time the new drugs are developed. Hence, it was reported that the private pharmaceutical industry also lacks interest in AMR research [27]. The following quote illustrates how long it takes to develop antibiotics and the high costs associated with their development:

“I think for private sectors, it’s not very encouraging…… Like once they develop [antibiotics], it took probably ten years to develop it until from the development until marketing they spend a lot of money. And then by the time it goes into the market, you probably will already see some type of resistance. Basically, they won’t be able to use it for a long time. And then this encouragement to not use it or like to use it at the minimum level. So it’s not very encouraging for private company to do that just because they won’t be making a hell lot of money, right?” (Participant 5)

A few participants reported that AMR research requires extensive studies and is complex due to their multifaceted nature. Participants highlighted that current funding streams are usually only available for a limited time, or for one-off program-specific or disease-specific AMR research uses. Further, funds available for a short period of time are inadequate for the gravity of the problem and are disproportionate to the AMR threat, urgency, and scale. The following quote illustrates that collaboration for AMR research requires long-term commitment and funds:

“So you have more and more funding, but sometimes it’s short. I mean for me, for example, three years funding it’s really short when you really want to, have an impact on something. This impact assessment as well of when you try to develop collaboration and so it’s most of the time missing because we start collaboration and you have a three years project and you have no money or no project or whatever to foresee in five years or ten years what’s happening to this collaboration.” (Participant 12)

Participants also mentioned that usually the funding available is limited to a specific organization or country. Researchers also described that transfer of funds to other nations was complicated even while being keen on collaborative work. Restriction of the use of funds within a country or organization limits the opportunity for collaboration in the region. The following two quotes illustrate the difficulties in moving funds across countries:

“The funding is not so flexible that we can use those funds overseas very easily. So there are a few fairly limited funds that we can use to do work outside of Singapore.” (Participant 2)

“I don’t see a whole lot of collaboration between universities. And I think that is because of the structure of the funding sometimes, you know? Yeah, structure of the funding, for example not being (transferable).” (Participant 5)

### 6.3. Different AMR Issues and Research Priorities in Asian Countries

Asia is vast and comprises both high to low-income nations with differing AMR issues. Studies have highlighted that, unlike HICs, many LMICs in the region have AMR problems stemming from improper or sub-optimal use of antibiotics. Antibiotics available in LMICs could be of inferior quality due to improper storage, and are also commonly available without a physician’s prescription [27]. In light of more pressing issue areas or limited resources, many low-income countries are also likely to prioritize other healthcare concerns over AMR [28]. It was reported that these diverse AMR challenges across contexts make collaboration tricky between countries in Asia. Here we can appreciate how researchers proposing different research protocols depending on the AMR problem in their country can impede intended joint ventures:

“We (are) like five countries. We have our own (need or) concept of the research… If our five countries, they have researcher team, the team leader and then they submit their proposal. They, they will have like a maybe in their research with it then there will be three or four or five sub project and that, and all of them will have what you say, go to the left basic style research.” (Participant 9) 

“On the other hand, the situation of AMR in Japan is not necessarily as severe as in some other places (in Asia).” (Participant 3)

Participants also reported that high income nations in Asia have better research capabilities in terms of laboratory facilities, manpower, and knowledge, as well as grants and funding than low-and-middle income countries. LMICs also lack resources for AMR stewardship, surveillance, and data management [29]. The following quotes emphasize how collaboration becomes difficult when there are gaps in knowledge and research capacity between stakeholders:

“I think that capacity is still quite a big issue in terms of having new labs that can do all the things to identify, you know, these resistant pathogens and having more coordinated surveillance mechanisms both nationally, I guess, and regionally. I think it’s still an issue. I think the capacity building is not necessarily very equitable across disciplines… I think it’s more difficult to build capacity in other areas of expertise that are needed.” (Participant 2)

“Because we have many microbiologist in Indonesia… now grad students… So the knowledge and skills seems still beginner area… So our target now and this month we have developed the education from advanced microbiology.” (Participant 6)

Participants also recognized the critical lack of data on antimicrobial consumption in different countries in Asia. This data is essential for important insights into the state of AMR in the region. The lack of such data presented a challenge during collaboration and ended up requiring a higher number of meetings and discussions between cross-country researchers for the initiation of collaborative research work.

“When you look through the data or the research in west, for example, in Sweden, in England and many countries in west that they publish a lot of articles regarding AMR antimicrobial consumption you easily find the easiest way to retrieve data and then to think about the pattern, but when we look at Asia, very difficult. And because of some in term of antimicrobial consumption because of via many countries all around Asia are different in term of law, in of prescription…And to measure the exact number or the exact consumption in human or in animals, that not easy. This is the first task that it’s a challenge for even people in our country as well to measure, but we have to convene many meetings.” (Participant 7)

### 6.4. Lack of Leadership and Asia-Centric AMR Institutions

It was reported that due to the absence of an Asia-centric institution for management of AMR, countries in the region adapted guidelines created by Europe or USA. Participants mentioned that there is a need for an institution based in Asia to facilitate and coordinate studies in the region to establish current AMR issues, data collection, and develop guidelines more contextualized to the needs of the region. Different geographical divisions created by Western organizations in Asia, such as the WHO regions, were cited to impede One Health collaborations between neighboring countries that happened to belong to different designated regions. The following quotes highlight the need for leadership initiated within Asia:

“I think it is a good idea if someone can be like a can be a center or facilitator or coordinator that have people from many countries in Asia and then working together.” (Participant 7)

“I don’t think that we have the Asian CDC. We don’t have service Asia CDC and you can see that the AMR in the Europe collaborate much better with the ECDC. In Africa you start to have Africa CDC to compile and, and be collaborative among more both research and the surveillance in the same pattern. And it can pull countries who are behind by the countries who are leaders. And I think the way that we actually or FAO-OIE split East Asia differently is a little bit difficult to form the network in the term of a One Health concept like the same way that ECDC do.” (Participant 4)

The lack of a regional governing institution has led to challenges in obtaining ethical approval studies. The requirements for multiple country-specific or organization-specific ethical approvals during the initiation of a collaboration between Asian countries have hindered or delayed collaborative research processes, thus discouraging the researchers in these endeavors.

“In Indonesia, we need to make the permission, like ethical review for each setting even though I already grant the permission from the… I might mention this, the hiring of the EC committee, ethical committee, because it’s from the Faculty of Indonesia and this is like the oldest committee for the EC that ever available in Indonesia even.” (Participant 10)

The participants also identified that a regional governing body could provide the much-needed guidelines and standardization of measures for the data collection on antimicrobial consumption, particularly for animal health in Asia. The following excerpt highlights the various methods the researchers attempted to collate data on antimicrobial consumption in farm animals in Asia: 

“We look together in among many research group and then we should what method to measure, but like for a livestock animals, it difficult to measure from farm to farm therefore we use like the association that have that like companies that sell antibiotic to farmers. We use that number of selling product, those farms and then calculate convert with that animal produced per year. We try to use daily dose and PCU. But that this time we still keep changing the unit of measurement. It’s not like a 100% that we use the European system, but we have to find our own system that suits us. And we believe that because of the difficulty in our country is maybe a good represent for the other country in Asia as well.” (Participant 7)

### 6.5. Impediment to Building Trust

Participants reported that trust among the different stakeholders is crucial for equitable collaborative work. Distrust can emerge among researchers from different nations or different organizations due to poor communication around the management of AMR projects and distinct levels of commitment to joint research studies. It was reported that equitable partnerships with transparent communication can build better trust among researchers. The following quote explains how the issue of trust led to the failure of a collaboration when the partnership was not equitable and mutually beneficial:

“So there’s, you know, a lot of collaborations are not possible or put on hold. And I think there’s also an issue of trust that, you know, when organizations or different institutions come together, you also need to build a good partnership, right? So an equitable partnership that it’s not like you are taking things from one party and so on. So it’s to be able to show interest and commitment to a particular place.” (Participant 15)

Furthermore, distrust among researchers prevailed when communication was poor and all collaborative partners were not made aware of the situation on the ground. The following quote illustrates how a HIC partner found it difficult to trust the transfer of funds for the work being conducted in neighboring LIMCs due to a lack of knowledge of the ongoing research activities: 

“Like, for example, in Singapore we have limited patient number. We need to go into the region because we are so affected by it, right? Just being a hub. Whatever is happening in Indonesia, in Malaysia, it’s going to impact us. But we don’t really know the situation there. So I think it’s an issue of trust. It’s an issue of how do you make funds flow between different countries.” (Participant 15)

Countries in Asia are culturally and linguistically diverse. At times, not sharing a common language makes communication difficult and complex. Language barrier was also identified as one of the challenges which can impede trust between collaborators as described in the next quote:

“I think, I mean there are I guess practical problems in terms of, you know, we do work in Cambodia. We don’t speak Malay so we have no idea what they’re saying. Or we’re doing, you know, we’re doing interviews of farmers, for example. We rely on them to do the interviews and then we have to wait for the translation to see what’s happened, right? It’s very hard to get a kind of real-time sense of what’s going on because we just don’t speak the language. So there are those kinds of issues.” (Participant 2)

### 6.6. Unequal Power Relationships

Participants reported that countries with higher technological capabilities or funding resources tended to assume the leadership role for AMR research work when cross-country collaboration occurs. Researchers in LMICs also hesitated to work with many stakeholders or other countries due to prior poor experiences of being denied recognition or credit for their contributions. The following quotes suggest that the researchers in LMICs were not keen for collaborations if they were expected to contribute only as support staff and did not benefit from the research work:

“Yeah, just do the groundwork and not really getting anything (recognition) out of it, right? You know there has been and it’s all this problem and we need to be careful about that as well, but there’s a lot more awareness about that now and I think even among those people who are researchers from outside the countries also, they are more wary about that. Just making us as data collectors and sample collectors.” (Participant 5)

The region receives a significant proportion of their funding from Western organizations. Challenges of collaboration also arose when the researchers were required to modify the study based on the funder’s specific requirements, which may not be the true representation of local research needs. Participants realized that an imbalance of power between the collaborators discourages the researchers from developing countries to collaborate with richer neighboring countries and the West.

“They (funders) will ask us as the team to modify the results as possible, you know, covered by the budget. So the modification is more to the whether it’s, how to say? Affordable or not. But not in detecting you need to measure this, measure that. You need to measure this way. Not in the time of methodology, but, yeah, the number of patients that will include it, what measurement that you want to take, how many time.” (Participant 10) 

### 6.7. COVID-19 Pandemic

AMR researchers also faced setbacks as the world was combating the pandemic. It was reported that health professionals, such as infection disease doctors, were roped into the treatment and management of COVID-19 patients as the number of COVID-19 patients soared across the region. Researchers faced challenges to obtain approval for new AMR research unless they were COVID-19-related studies. Ongoing AMR research projects were also paused as the researchers had to redirect their focus collectively to contribute to the public health effort on COVID-19 responses, or had limited capacity to continue ongoing AMR research due to COVID-19 restrictions. The following quotes exhibit the dilemma of the researchers in situations in which their AMR research was not approved or halted unless linked to a COVID-19 study:

“I mean like, we were able to propose (AMR research), but were not approved. So we have to move everything into COVID-related studies.” (Participant 5)

“COVID has affected us because it’s slowed down everything. Because a lot of the people who work on AMR, of course, now are working on COVID and it makes things difficult. But our organization’s mission is very AMR focused so that is what we do. If we do anything about COVID, it’s just to say like, for example, look what’s happened with the screening situation around COVID.” (Participant 3)

“Slow down and support is a little bit slow down because most of us are infectious diseases doctors, researchers in the COVID time we help with the COVID as well.” (Participant 4)

## 7. Opportunities

### 7.1. Presence of Willingness to Collaborate

Participants believed that researchers across Asia understood the importance and necessity of collaborative research, as AMR is not limited to specific borders. The willingness to work with neighboring countries showed promising future opportunities for Asia-centric research collaborations. Participants mentioned that they were indeed encouraged by their superiors to work with international agencies to establish collaborative work. The following quote highlight how institutions in Asia are pro-collaboration and have utilized the opportunity of attending conferences to network and form international partnerships: 

“One of the mission or the vision of our dean, to be able to collaborate more with international institutions. And we’ve been doing that. We’ve been having our faculty, encouraging our faculty to actually work with international agencies as well. And because of you know, sometimes when we attend certain conferences, you meet people. So that’s the way that we are able to start our collaborations.” (Participant 1)

One participant mentioned that collaborating with different organizations was a win-win scenario, where all involved parties benefit and learn from each other.

“I think that every organizations are quite open for the collaborations… whether it has to be a project, a study or a chatting over tea or discuss about the idea. And I think no matter in every level, I think that I feel that all are okay and they are open to talk with and everything is the standard of the world that when you want to work with the other group you have to go with discussion about what it can be the win-win scenario.” (Participant 4)

### 7.2. Networks for Collaborations

AMR researchers are building networks across Asia to establish efficient and effective communication. The development of multi-disciplinary networks has assisted in bringing AMR researchers from different fields together. Such networks also encouraged researchers to expand their knowledge by sharing their experiences and differing skills sets. Further, these networks fostered transparency in sharing research data and the utilization of funds between involved partners. Participants also mentioned that they understood the importance of personal connections to work in groups and form stronger professional relationships. The following quotes elucidate how AMR researchers in the region have been actively forming groups and networks to connect with other researchers to promote collaborations:

“Right now we the best that we can do is the connection, personal connection, then we group people working together, level of proposal and working together. But within Thailand we try to expand as much as we can to multi-disciplines of people networking together because the International Health Policy Program that they’re doing well to in order to gather people and then conducting or research together, yeah, they’re doing pretty well. Then they can do more in antimicrobial consumption.” (Participant 7)

“See, I mean… there are so many collaborators that, definitely, we do share the data with them and that’s one side of transparency is directly related to the data and data sharing. The other side is the finances and all those. So there are different aspects of transparency here. And for instance, while we do share the data with WHO, we’re already deploying also part of the class.” (Participant 8)

### 7.3. Prioritization of the One Health Paradigm

A One Health framework recognizes the interconnectedness of human health, agriculture, and animal health, together with the environment [28] and inspires researchers to address AMR, not only from a public health perspective, but also from historical, economic, and societal perspectives [30]. The application of a One Health framework promoted collaborative opportunities between researchers from different disciplines in the region. Further, the incorporation of a One Health approach has helped secure funds for collaborative research, especially from funders in the West, as they believe it to be an appropriate approach for combating AMR problems in the region. The following quotes illustrate the belief of AMR researchers that even a lay audience would understand the importance of the One health paradigm for AMR research:

“One Health is more or less understood by specialist or by lay audience, everybody acknowledges that it needs to be a joint effort and initiative while recognizing, and that was a slide that I revamped from many other more social sciences, health scientist or even journalist that address the problem at the narrative level.” (Participant 11)

“Ah yes, we basically most of the research that we conduct, we it based on the concept of One Health because we believe that a resistance pattern may be related between animals and humans and environment. Therefore we are doing more of our research this way.” (Participant 7)

“And then working as One Health approach for every sector and then we’re working together, that’s the reason why I got the second grant that working for AMR consumption, but in 270 hospitals then you can think of it.” (Participant 7)

## 8. Discussion

This study sought to understand the challenges and opportunities encountered by researchers while conducting collaborative AMR research in Asia. As documented in our results, we have identified a higher number of themes for challenges than opportunities for collaboration at present. Our study highlighted that the diverse nature of AMR, which requires cooperation between stakeholders across various One Health domains, the health sciences and non-health disciplines as well as private-public partnerships, presented immense challenges for researchers during research collaborations. Many researchers, owing to their educational background, work culture, or failure to understand the interconnectedness of the various health domains for AMR control, prefer to work with like-minded professionals from their own field. The lack of available funding and restrictions in transferring funds between stakeholders across organizations and countries prevented researchers from initiating combined efforts of AMR research. Funds not made available for the required length of time of research were also considered a hindrance to collaboration. Asia covers a huge geographical mass and comprises countries spanning high, middle, and low affluence levels. This region is also diverse in terms of economy, population, and social and cultural factors, leading to diverse AMR problems and research requirements. Researchers hence have struggled to develop flexible research agendas with shared priorities that are mutually beneficial to all involved parties. Although many international AMR institutes exist in the region, AMR researchers believed that Asia-centric leadership, institutions, and guidelines were lacking, hampering the promotion of much-required and appropriate Asia-focused AMR research. Our study further identified the impediment of trust among involved collaborators due to poor communication and commitment as an important challenge to improving future collaborations. The power imbalances among researchers from countries of different affluence where HICs tend to fund and lead while LMICs were expected to support has discouraged many researchers to commit to such ventures.

Lastly, COVID-19 had temporarily halted the AMR research in the region as healthcare resources, including AMR researchers, were diverted for the public health response to the pandemic. Approval for new AMR research and funds were also made scarce while the world tackled the different variants of SARS-CoV-2 infection.

AMR researchers’ appreciation of the need for combined efforts had led to positive inclinations towards equitable collaboration and provided the advancement of collaborative opportunities. Efforts on forming cross-discipline and cross-country networks have also brought researchers together for discussions on possible combinations of their individual studies and improved transparency in their communications. The acceptance and utilization of the One Health paradigm for research have promoted collaborative opportunities in Asia not only by providing a singular approach, but also by assisting the acquisition of vital funds for collaborative research.

It is globally acknowledged that AMR is an issue that transcends the medical discipline, with spill-over implications that severely affect society’s social, economic, and environmental realms [28]. Hence AMR research has to be vastly diverse and complex, ranging from clinical studies on the development of new treatment and diagnosis tools to behavioral and policy studies on the drivers behind AMR development across human, animal, and environmental health settings, as well as economic research measuring the cost-effectiveness of specific AMR interventions [2]. Hence, research, management, and mitigation of AMR require the contributions of professionals from different healthcare domains and the involvement of stakeholders and key decision-makers who are involved in representing and attaining society’s political and economic interests. One of the strategic objectives of the GAP-AMR is to “strengthen the knowledge and evidence base through surveillance and research” [2]. The GAP-AMR report cited the need for stakeholders, including national governments, international organizations, professional organizations, non-profits, industry and academia, to fill essential knowledge gaps on AMR [2]. Research agendas and priorities help guide overall research direction and promote the prudent allocation of resources on initiatives that are most necessary in moving the needle forward.

Although the United Nations (UN) declared AMR a major global health priority [31] and the WHO emphasized the need for the involvement of multi-sectoral stakeholders in AMR research [32], AMR remains a neglected domain of population health, especially in Asia. The AMR surveillance system lacks robustness especially in many LMICs in the region. The WHO’s global report on AMR surveillance in the year 2014 mentioned that only six of the eleven countries in the Southeast Asia had conducted AMR surveillance [30,33]. This could both be the cause and the consequence of the limited or lack of funds hindering the collaborative study within the region.

Asia is a vast continent comprising differing environmental, ecological, and economic factors affecting the rise of AMR [34]. The WHO Southeast Asia region alone is considered a “global hub for AMR emergence” as it runs the highest risk for AMR emergence among all WHO regions in Asia [7,10]. The development of AMR in LMICs is complex and may require mitigatory actions tailored to the specific need of these countries [28]. According to Ayukekbong et al., 2016, some of the factors driving AMR in LMICs are the inappropriate prescribing of antimicrobial drugs, poor patient awareness, limited diagnostic facilities, availability of antimicrobials without a prescription, lack of drug regulatory system, and inappropriate use of antimicrobials in animal production [27]. HICs in Asia have appropriate AMR surveillance and stewardship in place and are well equipped with expertise, technology, and funds to tackle their AMR problems, whereas LMICs in Asia lack such privileges [35]. Collaboration between countries with different affluence levels and AMR issues hence becomes difficult due to different research priorities and gaps in knowledge and research capacities.

Most AMR research in Asia is guided or funded by the institutions in the West, such as the Wellcome trust [36] and the Fleming Fund [37], which led to the establishment of collaborations between countries in Asia and the West, but less so between countries within the region. Collaborations in the region also get complicated by the impediment of trust and unequitable power relations between LMICs and HICs. The available AMR surveillance data in Asia are heavily heterogeneous and differ between countries [10], demonstrating the absence of cross-country coordination and communication. For example, although countries, such as India and China, are considered among the top five countries for antibiotic consumption, many LMICs in South and Southeast Asia lack data on their antibiotic use [35]. A key finding from the 2018 meeting in Singapore held to discuss the threat of AMR in Asia, which comprised experts across academia, industry, and government sectors from 10 countries, was the need for international collaboration in the region [7]. While citation and bibliometric analyses of the extent and scope of the AMR-related literature exist, and shed light on how these collaborations may be better facilitated or encouraged [38,39], there remains a dearth of information regarding AMR research coordination in Asia [30].

Since December 2019, the focus on the coronavirus disease 2019 (COVID-19) pandemic has been huge challenge for AMR researchers. AMR surveillance and antimicrobial stewardship (AMS) have been deprioritized as health system resources are being pooled for research, treatment, and management of the COVID-19 pandemic [40,41]. The researchers are also concerned about the probability of newly emerging AMR pathogens stemming from the overuse of broad-spectrum antibiotics to treat infections among COVID-19 patients [42,43]. Although our study participants recognized only the challenges brought by the pandemic, a survey conducted by Ashiru-Oredope et al. in 2021 in the United Kingdom highlighted a positive increase in multidisciplinary collaboration as well as an improved use of technology to identify the pathogens causing infections during the COVID-19 pandemic [44]. Moving forward, there is a critical need for prospective studies to understand the impact of COVID-19 on AMR and AMS, especially in the LMICs in the region [45,46].

Although the researchers have faced a multitude of challenges, AMR experts in the region understand the need for collaborative research to understand the current AMR situation and develop appropriate mitigation plans. Yam et al. (2019) acknowledged Singapore’s attempt at organizing a meeting between countries in Asia Pacific in the year 2018 to discuss the causes of AMR emergence, challenges of AMR surveillance, and possible mitigation processes [7]. This meeting also assisted in establishing networks for communication among the researchers involved. AMR researchers in Asia are also utilizing the One Health paradigm for their research which presents an opportunity to improve the collaborative efforts of multiple health science professions in the region [47].

It is imperative to tackle the above mentioned challenges to promote equitable collaborative AMR research planning in Asia. AMR researchers have recommended the establishment of an Asia-centric AMR institution. First, this institution can operate as a leading organization to craft AMR research agendas in Asia, support the formation of international research collaborations, and develop Asia-specific AMR guidelines. These guidelines can include recommendations for the prudent allocation of resources on initiatives that are most necessary in moving the needle forward. Second, this institution can serve as a regulatory body to eliminate the requirement for multiple ethical approvals and provide guidelines on antimicrobial consumption as well as standardize the unit of measures for data collection for such consumption. Third, the institution can serve as a platform to promote transparent communication [7]. The presence of this platform can improve trust among different stakeholders and enhance equitable collaborations in the region. Fourth, the institution can be responsible for assisting LMICs in improving their research capabilities. To this end, the support should include the enhancement of education and awareness of AMR, upgrades to laboratory capacity, the development and implementation of national action plans, infection prevention, and control guidelines. AMR researchers also desire equitable partnerships that are developed with trust and respect, in which HICs not only support capacity building in LMICs, but also acknowledge and share their contributions fairly. Doing so would diminish the reluctance of researchers in LMICs to collaborate due to prior experiences of power imbalance and distrust. The participation of the private sector in AMR research, especially in building advanced technology in disease diagnosis and the discovery of newer antibiotics, is also critical in collaborative work in the region. Public-private partnerships are also expected to present additional venues of much required funds for AMR research. Finally, collaborative AMR research in the region requires the understanding and agreement on the collective goals of a project, its joint planning and implementation, and early resolution of any conflicts.

## 9. Study Strengths and Limitations

To our knowledge, this is the first qualitative study investigating the challenges and opportunities of equitable collaborative AMR research in Asia. The participants recruited were senior researchers with ample AMR knowledge and research experiences in the region. The findings presented here are representative of the first-hand experiences encountered by participants during their course of collaborative research work. However, our findings should not be overgeneralized due to the purposive nature of the sample. During the COVID-19 pandemic, most AMR researchers were busy and unable to allocate time to participate in our study. Our participation rate hence was much less than initially expected, although we believe thematic saturation was reached as similar themes emerged from the interview data and these were rich in content. Data collection required a longer span of time and could only be conducted online due to the pandemic restrictions.

## 10. Policy Recommendations

Our study highlighted the need to improve equitable collaborative AMR research work in Asia and has identified the challenges and opportunities of such endeavors. Policy makers and researchers in Asia are committed to the development and implementation of NAP and emphasized collaborative research to improve the understanding of the causes and management of AMR problems [11]. Given that the challenges for collaboration arise due to the diverse nature of AMR and the involvement of multiple stakeholders from different human, animal, and environmental health and non-health professions, it becomes essential to develop new networks and improve existing networks between these stakeholders. A secure data-sharing platform can also enhance collaboration and trust among AMR experts. Policy makers in HICs are recommended to increase their awareness of neighboring LMICs, their research institutions, and AMR experts and pursue a collaborative partnership with them. Such collaborations should develop transparent and flexible agendas with shared leadership and research priorities [48,49]. LMICs are recommended to communicate their strengths and needs early on during the collaboration to avoid future conflicts [48]. It is also critical to understand and counter the power imbalances and distrust among collaborators from HICs and LIMCs. A mutually beneficial partnership, in which research contributions and findings from both HICs and LMICs are credited with fairness and equity, will promote trust and respect among collaborators, leading to increased opportunities for future joint ventures [49]. Further, researchers should be incentivized with rewards and/or recognition for their time and effort to construct equitable cross-discipline and cross-country collaborations. The institutions in Asia are also recommended to develop and lead their own research agendas in the region. To this end, there is a critical need for an Asia-centric regional governing body that serves a regulatory, capacity-building, and leadership function in managing AMR problems, as mentioned above. Finally, it is also essential to foster public-private partnerships between government institutions and for-profit organizations, such as private pharmaceutical companies, to enhance the much-needed collaborative AMR diagnostic and treatment research in Asia. Table 2 summarizes the policy recommendations for the challenges identified by our study.

## 11. Conclusions

While agreeable towards equitable, collaborative AMR research projects in the region, our participants outlined multiple challenges that hinder their efforts to work together. Regional AMR research is essential to provide improved and appropriate solutions to current AMR problems in Asia. More equitable collaborative partnerships could be developed by building networks across disciplines and countries in the region. An Asia-centric AMR institution to establish regional AMR leadership, and promote local collaborations, as well as support LMICs in addressing their identified needs, could be a step in the right direction.

## Figures and Tables

**Table 1 antibiotics-11-00755-t001:** Summary of the interview guide.

Summary Interview Guide
Research Priorities What do you think are the most pressing/urgent areas for AMR research in Asia? What research areas in AMR do you think should be prioritized in Asia right now? Challenges and Opportunities for Collaboration What do you think about the current state of collaboration between AMR researchers in the region? What were the major challenges that you faced during collaboration? What kind of support would you like to receive (from your home institution or otherwise) to better aid your research on AMR? Capacity Building for AMR Research in the Region How do you think capacity building on AMR research can be fostered in the region? How do you think academic partnerships can be developed to support researchers in conducting appropriate and equitable AMR research in the region? Understand the information flow mechanism in these networks and how it can be improved How do you exchange research information with your research collaborators? How do you think such information exchange can be improved? How do you share your research findings? Impact of COVID-19 on AMR research How has the COVID-19 pandemic impacted your work on AMR?

**Table 2 antibiotics-11-00755-t002:** Policy recommendations.

Challenges	Policy Recommendation
Multi-dimensional, multi-faceted nature of AMR	Develop multi-professional networks Build cross-country communication platforms Develop secure data-sharing platforms Foster public-private partnership
Funding Challenges	Ensure that donors respect the research priorities of countries and avoid imposing their research agendas Secure cross-country transferable funds Adequate funds made available for the required length of the research
Different AMR issues and research priorities in Asian countries	Communicate on the difference in research priorities and develop mutually beneficial research agendas Assist in capacity building for LMICs
Lack of Leadership and Asia-centric AMR institutions	Develop an Asia-centric regional governing body to oversee the AMR research in Asia
Impediment to trust	Develop mutually beneficial research agendas and processes Build transparent and flexible collaborations Increase communication among the researchers Reward and/or recognize researchers who construct equitable collaborations
Unequal power relationships	Develop mutually beneficial research agendas and processes Build research agendas with shared leadership Develop equitable partnerships with shared funding, shared decision making processes, and ensuring equal recognition of outputs Promote the new generation of researchers and make sure their work is recognized appropriately Reward and/or recognize researchers who develop equitable collaborations

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
