# Peer review of "Antimicrobial Resistance Research Collaborations in Asia: Challenges and Opportunities to Equitable Partnerships"

_antibiotics, 2022, doi:10.3390/antibiotics11060755_

Round 1

Reviewer 1 Report

This is a very interesting and an important study on the current condition of research collaborations of AMR researchers in Asia. As the authors highlight, AMR is a significant problem in Asia, more specifically in countries where antibiotic use is not strictly regulated. There is a serious need for more collaboration among countries to tackle the problem. 

Overall, the study was done well. I have the following questions and suggestions for the authors.

  1. How many countries in Asia did the respondents come from? This information was not clear from the text, and is important to understanding the diversity of the opinions presented.
  2. There is no need for the long second and third paragraphs in the results section. The authors simply listed all the subsections that follow in the results section in these paragraphs.
  3. I realize the authors are quoting the interviewed experts verbatim. However, there is no need to include sounds like "um" and "uh" in the text (lines 317 to 324).
  4. Few grammatical corrections: Line 118: there is no need for the word "be" before "fit". Line 606: please remove "a" before networks.  

Author Response

Comments

Response

This is a very interesting and an important study on the current condition of research collaborations of AMR researchers in Asia. As the authors highlight, AMR is a significant problem in Asia, more specifically in countries where antibiotic use is not strictly regulated. There is a serious need for more collaboration among countries to tackle the problem

Overall, the study was done well. I have the following questions and suggestions for the authors.

Thank you for your comments and recommendations.

How many countries in Asia did the respondents come from? This information was not clear from the text, and is important to understanding the diversity of the opinions presented.

Thank you for the question. We have added the countries in Asia represented by the participants in the result section in line 206. We did not collect other demographics due to privacy concerns. We hope this added information would suffice.

There is no need for the long second and third paragraphs in the results section. The authors simply listed all the subsections that follow in the results section in these paragraphs.

Thank you for the suggestion. We agree the two initial paragraphs in the results section summarize our findings, and we believe that these paragraphs provide an overview of our findings. If the reviewers are agreeable, we would like to keep this section as an introduction to our findings while we expand on the themes following this introduction part.

I realize the authors are quoting the interviewed experts verbatim. However, there is no need to include sounds like "um" and "uh" in the text (lines 317 to 324).

We have reviewed all the excerpts and removed the sounds like “um” and “uh” recorded in the transcription.

Few grammatical corrections: Line 118: there is no need for the word "be" before "fit". Line 606: please remove "a" before networks. 

Thank you for catching the grammatical errors. We have done a grammar check for the manuscript and made the required corrections.

Reviewer 2 Report

I have a few comments that may help improve the quality of the Manuscript:

Minor English language edits i.e. There has been calls 

In the introduction section, can a comparison of key differences among the countries in handling be described in greater detail, rather than just descriptively? 

Can some examples be included on how the incoherence resulted in outbreaks of multiresistant organisms across countries?

Was the study approved by an ethics committee?

How was the question guide designed?

Sampling subjects from a pool of authors of the same paper may have resulted in bias, please comment or include in the limitations of your study.

In challenges, farmers are not mentioned but still may be great contributors to amr. Can something be said about this as well?

Also, exact effects of COVID-19 on amr can be described in more detail as not all of them are negative.

Author Response

Comments

Response

Minor English language edits i.e. There has been calls

Thank you for catching the grammatical error. We have done a grammar check for the manuscript and made the required corrections.

In the introduction section, can a comparison of key differences among the countries in handling be described in greater detail, rather than just descriptively?

Thank you for the suggestion. We have added examples of discrepancies between countries in Asia while implementing policies on the use of antibiotics in animal farming. This information is added in lines: 105-114.

Can some examples be included on how the incoherence resulted in outbreaks of multiresistant organisms across countries?

Thank you again for your suggestion. There is a dearth of literature available on outbreaks of multi-resistant organisms due to incoherence in policy, implementation, and AMR research between the countries in Asia. However, we have added specific examples of pathogens being identified across borders in the region. This information is added in lines: 57-66.

Was the study approved by an ethics committee?

The study was submitted to National University Singapore (NUS) Institutional Review Board (IRB) for Social, Behavioural, and Educational Research (SBER) for approval. This board provided an exemption from ethical requirements for this study.

How was the question guide designed?

Thank you for the question. We have added an excerpt detailing how our question guide was developed in the method section on lines: 138-142.

Sampling subjects from a pool of authors of the same paper may have resulted in bias, please comment or include in the limitations of your study.

Thank you for your comment and your recommendation. We have included the potential sampling bias resulting from purposeful sampling of authors in our study limitation in lines: 690.

In challenges, farmers are not mentioned but still may be great contributors to amr. Can something be said about this as well?

Thank you for the recommendation. While meaningful, collaboration with farmers would imply collaboration at the more local level, we are focused on looking at collaboration between nations at the international/regional, i.e. more macro level. Hence the challenges relating to the farmers’ contribution to AMR are not relevant to our study. We hope our justification for not including this suggestion is acceptable to you.

Also, exact effects of COVID-19 on amr can be described in more detail as not all of them are negative.

Thank you for the suggestion. To provide a holistic picture of the effect of Covid-19 pandemic on AMR research, we have included the opportunities brought by the Covid-19, in AMR research collaboration in the discussion section on lines: 643-646.